# *Slowpoke* functions in circadian output cells to regulate rest:activity rhythms

**Daniela Ruiz, Saffia T. Bajwa, Naisarg Vanani, Tanvir A. Bajwa, Daniel J. Cavanaugh***

Department of Biology, Loyola University Chicago, Chicago, Illinois, United States of America

* dcavanaugh1@luc.edu

**Data Availability Statement:** All scRNAseq files are available from the Gene Expression Omnibus database (accession number GSE162379).

**Funding:** This work was supported by the National Institute of General Medical Sciences,

## Abstract

The circadian system produces ~24-hr oscillations in behavioral and physiological processes to ensure that they occur at optimal times of day and in the correct temporal order. At its core, the circadian system is composed of dedicated central clock neurons that keep time through a cell-autonomous molecular clock. To produce rhythmic behaviors, time-of-day information generated by clock neurons must be transmitted across output pathways to regulate the downstream neuronal populations that control the relevant behaviors. An understanding of the manner through which the circadian system enacts behavioral rhythms therefore requires the identification of the cells and molecules that make up the output pathways. To that end, we recently characterized the *Drosophila* pars intercerebralis (PI) as a major circadian output center that lies downstream of central clock neurons in a circuit controlling rest:activity rhythms. We have conducted single-cell RNA sequencing (scRNAseq) to identify potential circadian output genes expressed by PI cells, and used cell-specific RNA interference (RNAi) to knock down expression of ~40 of these candidate genes selectively within subsets of PI cells. We demonstrate that knockdown of the *slowpoke* (*slo*) potassium channel in PI cells reliably decreases circadian rest:activity rhythm strength. Interestingly, *slo* mutants have previously been shown to have aberrant rest:activity rhythms, in part due to a necessary function of *slo* within central clock cells. However, rescue of *slo* in all clock cells does not fully reestablish behavioral rhythms, indicating that expression in non-clock neurons is also necessary. Our results demonstrate that *slo* exerts its effects in multiple components of the circadian circuit, including PI output cells in addition to clock neurons, and we hypothesize that it does so by contributing to the generation of daily neuronal activity rhythms that allow for the propagation of circadian information throughout output circuits.

## Introduction

Behavioral circadian rhythms depend on dedicated clock neurons in the brain that track time of day through the function of a molecular circadian clock. In the fruit fly, *Drosophila melanogaster*, there are ~150 central clock neurons in the brain, as determined by expression of components of the molecular clock. These clock neurons include the large and small ventral lateral

www.nigms.nih.gov/, Grant R15GM128170 to D.J. C, and the Brain and Behavior Research Foundation, https://www.bbrfoundation.org/, Young Investigator Grant #24045 to D.J.C. The funders had no role in study design, data collection and analysis, decision to publish, or preparation of the manuscript.

**Competing interests:** The authors have declared that no competing interests exist.

neurons (lLNv and sLNv, respectively), the dorsal lateral neurons (LNd), the lateral posterior neurons (LPN), and three groups of dorsal neurons (DN1, DN2 and DN3) [1]. In addition to clock cells in the brain, molecular clocks have been identified in numerous peripheral tissues, where they are thought to regulate circadian control of tissue-specific functions [2, 3].

It is hypothesized that central clock neurons modulate behavior through neuronal connectivity with downstream output regions. Thus, an understanding of circadian control of behavior necessitates identification of output cell populations. We recently demonstrated that the pars intercerebralis (PI), functional equivalent of the mammalian hypothalamus, comprises a major circadian output center in *Drosophila* [4, 5]. The PI can be divided into several distinct neuronal subtypes that differ in terms of neuropeptide expression, projection patterns, and function [6]. Interestingly, these subtypes contribute differentially to circadian control of behavior and physiology. PI neurons that express the neuropeptide SIFamide (SIFa) project broadly throughout the brain and ventral nerve cord [5, 7, 8], and manipulations of these cells affect circadian rest:activity and feeding:fasting rhythms [4, 5]. A distinct subset expressing the neuropeptide diuretic hormone 44 (DH44), a homolog of the mammalian corticotropin-releasing factor, has a more circumscribed projection pattern [5, 6] and appears to selectively regulate rest:activity but not feeding:fasting rhythms [5, 9]. Finally, a third subset known as the insulin-producing cells (IPCs), which is defined by expression of the *Drosophila* insulin-like peptides (DILPs), is dispensable for both rest:activity and feeding:fasting rhythms [4, 5], and may instead mediate interactions between central and peripheral clock tissues [10].

A major question is how circadian information generated by clock cells is conveyed across output circuits to ultimately control behavioral and physiological processes. Because PI cells lack molecular clocks, their ability to transmit circadian information likely relies on cyclic inputs from central clock cells. Consistent with this idea, PI output cells have been shown to receive synaptic inputs from clock neurons [5, 10]. In flies and mammals, central clock neurons exhibit rhythms of cell excitability that result from oscillations in gene expression under control of the molecular clock [11–17], thus translating the ticking of the molecular clock into cyclic neuronal outputs. More recently, several groups have reported oscillations in neuronal activity in multiple putative circadian output cell populations in *Drosophila*, including DH44-expressing PI cells [10, 18–20]. Notably, these oscillations are under control of the central brain clock [19, 20], which supports a model in which neuronal activity rhythms are first generated in clock cells via molecular clock mechanisms and then propagated to downstream output cells to impose circadian modulation on behavioral processes. This model is consistent with the fact that constitutive activation or inhibition of different output cell populations, which would abrogate clock-driven neuronal activity rhythms in these cells, drastically decreases rest:activity rhythm strength [4, 5, 9, 19].

In addition to pinpointing output cell populations and tracing circuits controlling distinct behaviors, it is also essential to understand the molecular mechanisms that confer upon output cells the ability to transmit circadian information. This includes identification of circadian output genes that contribute to circadian rhythms without affecting molecular clock function. To date, few circadian output genes have been linked to regulation of behavioral rhythms, and most of these act within core clock neurons themselves, rather than downstream output areas [21, 22]. Identification of output genes will enhance our understanding of the function of output circuits, and furthermore will provide insight into the health consequences associated with circadian disruption and aging. For example, the fragmentation of sleep-wake cycles that occurs with aging is associated with a gradual reduction in circadian rhythm strength that results in part from decreased coupling between the central clock and output pathways [23, 24]. Output genes likely play an important role in this process.

Here, we sought to identify circadian output genes that function in PI cells to control behavioral rest:activity rhythms. We predicted that the ability of PI cells to propagate circadian information depends on the expression in output cells of 1) receptors for neuropeptides and neurotransmitters that are released from central clock cells, 2) ion channels and intracellular signaling molecules that regulate neuronal excitability, and 3) neuropeptides and neurotransmitters that are released from output neurons to communicate with downstream components of the output circuit. We therefore performed scRNAseq to identify candidate neuronal signaling molecules expressed by PI output cells, and conducted behavioral rest:activity monitoring following PI-specificRNAi-mediated knockdown of these molecules. Through these experiments, we identify a role for the *slowpoke* potassium channel in specific PI cell subsets as a critical regulator of circadian rest:activity outputs.

## Materials and methods

### Fly lines

We ordered the following fly lines from the Bloomington Drosophila Stock Center (BDSC): C767-GAL4 (RRID:BDSC_30848), UAS-Dicer2 (RRID:BDSC_24650 and RRID: BDSC_24651), UAS-nlsGFP (RRID:BDSC_7032), UAS-mCD8::GFP (RRID:BDSC_5130), and DILP2-GAL4 (RRID:BDSC_37516). We ordered DH44-GAL4 (VT ID 039046) from the Vienna Drosophila Resource Center (VDRC) [25]. SIFa-GAL4 [8], kurs58-GAL4 (FBti0017957) [26] and Dilp2mCherry (FBti0202307) [5] were gifts from Amita Sehgal. C929-GAL4 (FBti0004282) [27] was a gift from Paul Taghert. We obtained RNAi lines for behavioral screening from the VDRC and the BDSC (see S1 File for a complete list of RNAi lines) [28, 29].

### Single-cell RNA sequencing

We used a single-cell transcriptional profiling approach to identify potential circadian output genes expressed by relevant PI cell populations. The PI is comprised of ~30 cells, but only specific subsets have been implicated in control of rest:activity rhythms. Because the 14 DILP-expressing PI cells do not appear to contribute to rest:activity regulation [4, 5], we sought to target non-DILP-expressing PI cells for single-cell sequencing following GFP-guided cell capture. To identify the cells of interest, we drove GFP expression with either of two GAL4 lines, kurs58-GAL4 or C767-GAL4, which are both active in non-DILP-expressing PI cells [5]. Notably, constitutive neuronal activation under the control of either kurs58-GAL4 or C767-GAL4 compromises rest:activity rhythm strength, confirming the relevance of these cells [5]. The flies used for single-cell capture also included a Dilp2mCherry construct, which selectively labels the DILP-expressing PI cells. This served two purposes: first, Dilp2mCherry acted as a landmark to aid in PI localization; second, it allowed us avoid selecting DILP-expressing cells, which could be easily identified based on their mCherry fluorescence (see Fig 1A–1C).

We performed single-cell sequencing analysis on a total of 5 PI cells that were labeled by either kurs58-GAL4 or C767-GAL4. Following analysis, one cell was excluded due to indiscriminate mapping of reads. Kurs58-GAL4/Dilp2mCherry; UAS-mCD8::GFP/+ or UAS-nlsGFP /Dilp2mCherry; c767-GAL4/+ flies were anesthetized briefly with $CO_2$, glued down onto a 35mm tissue culture dish (Falcon), and head cuticle was dissected off to expose the brain. Flies were submerged in HL3.1 [30] during cell harvesting. PI cells were visualized with an inverted microscope (Olympus BX61WI) with LUMPlanFl immersion objectives (20 x /0.50W and 40 x /0.80W). Since other cell types, in addition to PI cells, are labeled by GFP in both kurs58-GAL4 and C767-GAL4 flies, we used mCherry fluorescence to locate the PI, and only selected GFP-expressing cells adjacent to mCherry cells.

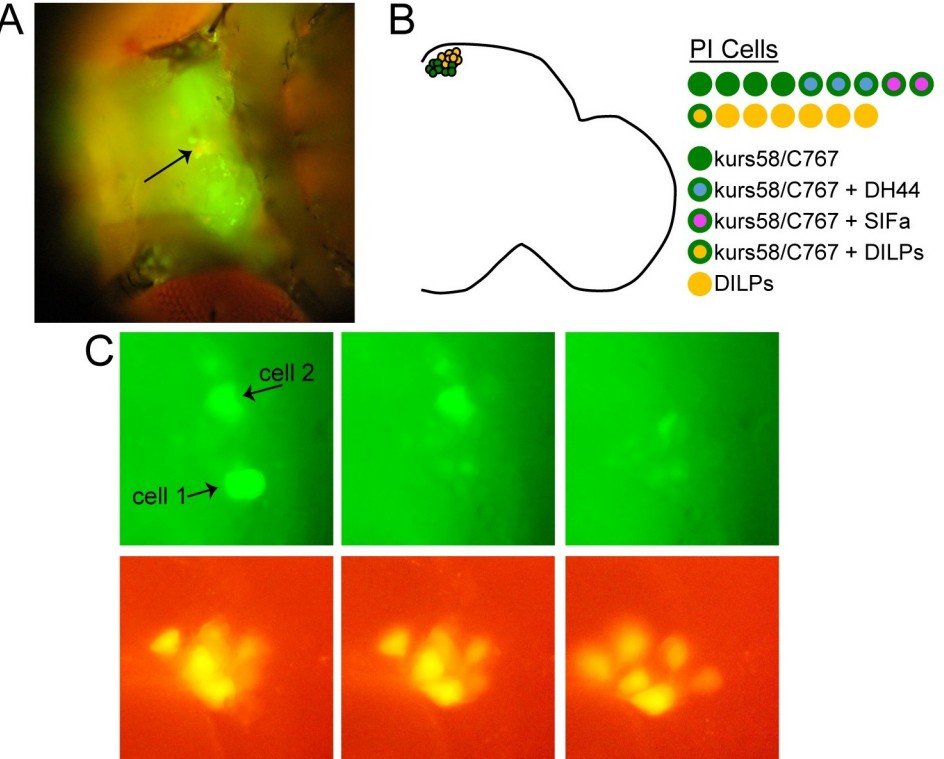

**Fig 1. Single PI cell harvesting for RNA sequencing analysis.** (A) Fluorescence microscopy image of a UAS-nlsGFP /Dilp2mCherry; C767-GAL4/+ fly prepared for PI cell harvesting. Head cuticle between eyes has been removed, revealing the dorsal surface of the brain. Arrow points to PI region. (B) Schematic showing different PI cell types. On the left, one hemisphere of the fly brain is depicted. kurs58-GAL4 and C767-GAL4 (green circles) are largely restricted to the non-DILP-expressing PI neurons. DILP-expressing neurons are depicted in orange. The neurochemical makeup of the different PI cells is detailed on the right. In each hemisphere, there are ~7 DILP-expressing PI cells (orange cirlces) and ~10 kurs58/C767-GAL4-expressing PI cells (green circles). The neuropeptides DH44 and SIFa are present in non-overlapping populations of the kurs58/C767-GAL4-expressing cells. 3 of these cells express DH44 (green circles with blue interior); 2 express SIFa (green circles with magenta interior). (C) Closeup of the PI region showing two GFP-expressing PI cells (top) that were harvested for single-cell sequencing. Sequential images were taken before and after harvesting each cell. Nearby Dilp2mCherry-expressing cells were unaffected (bottom).

We used a fine-tipped glass micropipette for cell harvesting. The micropipette was inserted into a pipet holder and connected by flexible tubing to a 1 mL syringe. Using a micromanipulator, we slowly advanced the micropipette towards the PI region. To avoid collecting cellular debris while advancing through brain tissue, we maintained light positive pressure by blowing through the syringe. Once the micropipette was just touching the soma of the cell of interest, we applied gentle mouth suction until the cell entered the pipet tip. We then broke off the tip containing the harvested cell into a 1.7 mL microcentrifuge tube and immediately processed the contents for antisense RNA amplification.

Single-cell RNA was processed through three rounds of antisense RNA amplification [31], and libraries were made from the amplified material using Illumina Truseq v2 reagents. 100 base pair, single-end RNA sequencing was performed by the Institute for Diabetes, Obesity and Metabolism Functional Genomics Core at the University of Pennsylvania. Approximately 10 million raw reads were obtained per sample. PolyA, adapter and low quality sequences were trimmed with Trim Galore (https://github.com/FelixKrueger/TrimGalore) and PRINSEQ [32]. Following filtering >90.42% of reads remained for each sample.

Reads were mapped to Dmel Reference Genome (dm6) with RNA Star [33] in Galaxy (using the public Galaxy server at usegalaxy.org). Drosophila_melanogaster.BDGP6.87.gtf was used for the annotation file. ~60–85% of reads were uniquely mapped across all samples. Mapped reads were normalized to reads per million (RPM) for each sample (based on total unique mapped reads that could be unambiguously assigned to a gene), and log2 transformed (RPM +1). For identification of candidate signaling genes, we defined significant expression as $\geq 1$ RPM. This relatively low threshold reflects the fact that many signaling genes, in particular receptor molecules, are expressed at comparatively low levels [34].

Because the aRNA amplification process results in extreme 3' bias of reads (due to the use of oligo(dT) primers for first-strand cDNA synthesis) and because our libraries were unstranded, mapping was ambiguous if genes on different strands had overlapping 3' ends. Ambiguous reads are not assigned to either gene, resulting in potential underestimate of abundance of such genes. We therefore manually inspected each gene of interest for 3' overlap, and additionally used MMQuant [35] to identify reads that ambiguously mapped to multiple genes. Tables 1–3 indicate any potential underestimation of gene expression by italicizing gene expression values of genes for which there were ambiguously mapped reads due to 3' overlap.

## Rest:Activity rhythm analysis

Flies were raised on cornmeal-molasses medium and were entrained to a 12:12 Light-Dark (LD) cycle at 25˚C prior to behavioral experiments. Following entrainment, individual ~7 d old male flies were loaded into glass tubes containing 5% sucrose and 2% agar for locomotor activity analysis with the *Drosophila* Activity Monitoring (DAM) System (Trikinetics, Waltham MA). DAMS monitoring was conducted at 25˚C in constant dark (DD) conditions and data were acquired every minute. For each individual fly, rest:activity rhythm period and strength (power) were determined for days 2–7 of DD with ClockLab software (Actimetrics, Wilmette IL) using chi-square periodogram analysis. Rhythm power was calculated as the amplitude of the periodogram line at the dominant period minus the chi-square significance line (at a significance of $p < 0.01$). Flies that died during the course of behavioral monitoring were identified via visual inspection of activity records and removed from analysis. All flies that survived through the end of the one-week monitoring period were included in mean rest: activity power determination. Because rhythm strength cannot be negative, flies with a calculated power $< 0$ were assigned a power of 0 for subsequent analysis. Representative individual activity records displayed in figures were selected to have a rhythm power that fell within the 95% confidence interval of the mean for a given genotype. Only rhythmic flies (defined as a power $> 100$), were included in period estimation.

The SIFa/DH44-GAL4 line used for behavioral screening contains a combination of SIFa-GAL4 and DH44-GAL4 to drive RNAi expression selectively in 10 cells of the PI, and also includes UAS-Dicer2 to increase RNAi efficiency. The full genotype of this line is SIFa-GAL4, UAS-Dicer2; DH44-GAL4. All components of this line were created in or outcrossed $\geq 5$ times to the iso[31] (isogenic w[1118]) stock [36]. In our initial screen, SIFa/DH44-GAL4 flies were crossed to UAS-RNAi flies to create experimental lines. Controls consisted of the DH44/SIFa-GAL4 line crossed to the iso[31] stock. We conducted two independent behavioral experiments for each experimental line (each experiment with ~16 flies per genotype), and pooled results for analysis. Because GAL4 control flies were run alongside experimental lines in each run, the *n* for this group is substantially larger than that of the experimental groups. For retests with the *sss* and *slo* RNAi lines, we conducted ~5 independent behavioral experiments for each genotype (each experiment with ~16 flies per genotype), and pooled results for analysis. In rescreen experiments, we compared each experimental to two genetic controls: in addition to the GAL4

**Table 1. Circadian receptor gene expression in PI output cells.**

| Gene Name | Flybase Gene Number (FBgn) | Cell 1 | Cell 2 | Cell 3 | Cell 4 | Tested in RNAi screen? |
|---|---|---|---|---|---|---|
| --Acetylcholine Receptors-- | | | | | | |
| nicotinic Acetylcholine Receptor α1 (96Aa) | FBgn0000036 | 0.00 | 6.16 | 4.87 | 1.37 | x |
| nicotinic Acetylcholine Receptor α2 (96Ab) | FBgn0000039 | 0.00 | 2.64 | 0.00 | 0.00 | |
| nicotinic Acetylcholine Receptor α3 (7E)* | FBgn0015519 | 0.00 | *3.19* | *7.02* | *6.55* | x |
| nicotinic Acetylcholine Receptor α4 (80b) | FBgn0266347 | 0.33 | 4.45 | 0.32 | 5.27 | x |
| nicotinic Acetylcholine Receptor α5 (34E) | FBgn0028875 | 1.19 | 3.95 | 2.23 | 4.46 | x |
| nicotinic Acetylcholine Receptor α6 (30d) | FBgn0032151 | 7.82 | 5.75 | 6.02 | 6.82 | x |
| nicotinic Acetylcholine Receptor α7 (18C) | FBgn0086778 | 7.07 | 6.63 | 0.32 | 4.97 | x |
| nicotinic Acetylcholine Receptor β1 (64B) | FBgn0000038 | 0.33 | 3.78 | 8.99 | 5.18 | x |
| nicotinic Acetylcholine Receptor β2 (96A) | FBgn0004118 | 0.00 | 0.00 | 0.00 | 0.00 | |
| nicotinic Acetylcholine Receptor β3 (21C)* | FBgn0031261 | *0.00* | 0.00 | 0.00 | *0.00* | |
| muscarinic Acetylcholine Receptor, A-type (60C)* | FBgn0000037 | 0.00 | 0.37 | *0.00* | 4.07 | |
| muscarinic Acetylcholine Receptor, B-type | FBgn0037546 | 0.33 | 0.00 | 0.32 | 0.00 | |
| muscarinic Acetylcholine Receptor, C-type | FBgn0029909 | 0.00 | 0.00 | 0.00 | 0.00 | |
| RIC3 acetylcholine receptor chaperone | FBgn0050296 | 0.60 | 0.90 | 0.00 | 0.54 | |
| --Glutamate Receptors-- | | | | | | |
| DmGluRA | FBgn0019985 | 0.33 | 0.00 | 0.00 | 5.12 | |
| dNR1/ NMDAR-I | FBgn0010399 | 0.00 | 0.37 | 0.00 | 0.00 | |
| dNR2/ NMDAR-II | FBgn0053513 | 0.00 | 0.37 | 0.00 | 0.00 | |
| GluRI/GluRIA | FBgn0004619 | 0.00 | 0.00 | 3.81 | 1.23 | |
| GluRIB | FBgn0264000 | 0.00 | 1.60 | 0.00 | 4.41 | |
| GluRIIA | FBgn0004620 | 0.33 | 0.00 | 0.00 | 0.00 | |
| GluRIIB | FBgn0020429 | 0.00 | 0.00 | 0.00 | 0.00 | |
| GluRIIC/GluRIII | FBgn0046113 | 0.00 | 0.00 | 0.00 | 0.00 | |
| GluRIID | FBgn0028422 | 0.00 | 0.00 | 0.00 | 0.00 | |
| GluRIIE | FBgn0051201 | 0.00 | 3.04 | 0.00 | 4.14 | x |
| clumsy/GluR39B | FBgn0026255 | 0.00 | 0.00 | 0.00 | 0.00 | |
| KaiR1C/Grik | FBgn0038840 | 5.04 | 0.37 | 0.00 | 0.00 | |
| KaiR1D | FBgn0038837 | 0.00 | 0.00 | 0.00 | 0.00 | |
| CG11155 | FBgn0039927 | 0.60 | 5.29 | 6.29 | 7.12 | x |
| Ekar/CG9935 | FBgn0039916 | 0.33 | 3.14 | 0.00 | 5.78 | x |
| GluCl | FBgn0024963 | 0.33 | 2.88 | 9.66 | 2.52 | x |
| Neto | FBgn0265416 | 0.33 | 0.90 | 0.00 | 5.88 | |
| Nmda1* | FBgn0013305 | *7.70* | *5.55* | *9.63* | *5.72* | x |
| --Glycine Receptors-- | | | | | | |
| Grd* | FBgn0001134 | 0.00 | *1.73* | 0.00 | *0.00* | |
| CG12344* | FBgn0033558 | 0.00 | *2.88* | 0.00 | *0.00* | |
| CG7589 | FBgn0036727 | 0.00 | 6.71 | 0.00 | 5.36 | |
| --Peptide Receptors-- | | | | | | |
| AstC-R1 | FBgn0036790 | 0.00 | 3.51 | 0.00 | 0.74 | |
| AstC-R2 | FBgn0036789 | 0.00 | 0.66 | 0.00 | 0.00 | |
| CCHa1-R | FBgn0050106 | 0.00 | 0.00 | 0.00 | 0.29 | |
| CCHa2-R | FBgn0033058 | 0.33 | 0.66 | 8.54 | 0.29 | |
| Dh31-R1 | FBgn0052843 | 0.00 | 2.94 | 0.00 | 0.00 | |
| Dh44-R1 | FBgn0033932 | 0.00 | 0.00 | 0.00 | 0.00 | |
| Dh44-R2* | FBgn0033744 | 0.00 | *5.11* | *0.00* | *0.00* | |
| Lkr | FBgn0035610 | 0.00 | 0.00 | 0.00 | 0.00 | |

*(Continued)*

**Table 1.** (*Continued*)

| Gene Name | Flybase Gene Number (FBgn) | Cell 1 | Cell 2 | Cell 3 | Cell 4 | Tested in RNAi screen? |
|---|---|---|---|---|---|---|
| NPFR | FBgn0037408 | 0.00 | 0.00 | 0.32 | 0.00 | |
| Pdfr | FBgn0260753 | 6.92 | 5.76 | 0.00 | 6.06 | x |
| PK2-R1 | FBgn0038140 | 0.00 | 0.00 | 0.00 | 0.00 | |
| PK2-R2 | FBgn0038139 | 0.00 | 0.00 | 0.00 | 0.00 | |
| SIFaR | FBgn0038880 | 0.00 | 0.00 | 0.00 | 0.00 | |
| sNPF-R | FBgn0036934 | 2.19 | 0.00 | 7.33 | 0.00 | x |

Numbers are log2-transformed reads per million.

* indicates mapping ambiguity due to overlapping 3' region. Italics indicates a potential underestimation of actual read number due to this ambiguity. Shading indicates relative gene expression level, with hotter colors representing higher expression levels.

control used in the initial screen, we also included a UAS control, which consisted of each UAS-RNAi line crossed to the iso[31] stock.

For other behavioral experiments, we assessed the effect of RNAi-mediated knockdown in subsets of PI cells using the c929-GAL4 and DILP2-GAL4 drivers. These GAL4 lines were combined with a third chromosome UAS-Dicer2 line to create the following stocks: c929-GAL4; UAS-Dicer2; and DILP2-GAL4; UAS-Dicer2. The resultant stocks were crossed with UAS-RNAi lines to create experimental flies that were compared with GAL4 and UAS controls, which consisted of the GAL4 or UAS-RNAi lines crossed to the iso[31] stock. For these experiments, we ran 2–4 independent behavioral experiments for each genotype (each experiment with ~16 flies per genotype), and pooled results for analysis.

## Immunohistochemistry

Adult (~7d old) fly brains were dissected in phosphate-buffered saline with 0.1% Triton-X (PBST) and fixed in 4% formaldehyde for 20–35 min. Brains were rinsed 3 X 15 min with PBST, blocked for 60 min in 5% normal donkey serum in PBST (NDST), and incubated for 24 hrs at RT in rabbit anti-GFP (Molecular Probes A-11122) diluted 1:1000 in NDST. Brains were then rinsed 3 X 15 min in PBST, incubated for 24 hrs in FITC donkey anti-rabbit (Jackson 711-095-152) diluted 1:1000 in NDST, rinsed 3 X 15 min in PBST, cleared for 5 min in 50% glycerol in PBST, and mounted in Vectashield. Immunolabeled brains were visualized with a Fluoview 1000 confocal microscope (Olympus).

## Statistical analysis

Statistical analysis was performed with GraphPad Prism 8.4.3 software (La Jolla, CA). One-way ANOVA with Dunnett's multiple comparisons test was used to compare each experimental line to the common GAL4 control in our initial screen. One-way ANOVA with Tukey's multiple comparisons test was used to compare each experimental line with both GAL4 and UAS controls in subsequent behavioral experiments. For all analyses, $p < 0.05$ was considered significant.

## Results

### PI cell expression of receptors for neuropeptides and small molecular neurotransmitters implicated in circadian rhythm regulation

We previously demonstrated that non-DILP-expressing PI cells comprise essential components of a circadian output circuit controlling rest:activity rhythms [4, 5]. To better understand

**Table 2. Non-circadian receptor gene expression in PI output cells.**

| Gene Name | Flybase Gene Number (FBgn) | Cell 1 | Cell 2 | Cell 3 | Cell 4 | Tested in RNAi Screen? |
|---|---|---|---|---|---|---|
| --Dopamine Receptors-- | | | | | | |
| Dop1R1 | FBgn0011582 | 4.76 | 4.60 | 0.32 | 0.00 | |
| Dop1R2 | FBgn0266137 | 0.33 | 0.00 | 0.58 | 0.00 | |
| Dop2R | FBgn0053517 | 8.19 | 3.87 | 6.96 | 4.77 | x |
| DopEcR | FBgn0035538 | 1.48 | 7.02 | 1.89 | 8.18 | x |
| --GABA Receptors-- | | | | | | |
| Rdl | FBgn0004244 | 7.41 | 5.80 | 9.51 | 6.74 | x |
| GABA-B-R1 | FBgn0260446 | 0.00 | 0.37 | 3.06 | 3.67 | |
| GABA-B-R2 | FBgn0027575 | 0.33 | 0.00 | 7.15 | 0.00 | |
| GABA-B-R3 | FBgn0031275 | 0.00 | 0.00 | 0.00 | 0.00 | |
| --Histamine Receptors-- | | | | | | |
| HisCl1 | FBgn0037950 | 0.00 | 0.00 | 0.00 | 5.30 | |
| ort | FBgn0003011 | 0.00 | 0.00 | 0.00 | 0.00 | |
| --Octopamine Receptors-- | | | | | | |
| oamb | FBgn0024944 | 0.33 | 4.77 | 0.00 | 0.29 | |
| Octβ1R | FBgn0038980 | 0.00 | 0.00 | 1.15 | 4.86 | |
| Octβ2R | FBgn0038063 | 5.98 | 6.89 | 7.31 | 7.95 | x |
| Octβ3R | FBgn0250910 | 0.00 | 0.00 | 0.00 | 2.13 | |
| Oct-TyrR | FBgn0004514 | 0.00 | 0.00 | 0.80 | 0.00 | |
| Octα2R | FBgn0038653 | 3.20 | 6.71 | 5.10 | 6.33 | |
| --Peptide Receptors-- | | | | | | |
| AdoR* | FBgn0039747 | 0.00 | *3.72* | 0.00 | 0.93 | |
| AkhR | FBgn0025595 | 8.33 | 8.49 | 5.05 | 6.31 | x |
| AstA-R1 | FBgn0266429 | 0.33 | 0.37 | 0.32 | 3.35 | |
| AstA-R2 | FBgn0039595 | 0.00 | 0.00 | 7.05 | 0.29 | |
| capaR | FBgn0037100 | 0.00 | 0.00 | 0.00 | 0.00 | |
| CCAP-R | FBgn0039396 | 0.00 | 0.37 | 5.24 | 0.00 | |
| CCKLR-17D1 | FBgn0259231 | 0.33 | 3.87 | 4.54 | 0.00 | |
| CCKLR-17D3 | FBgn0030954 | 0.00 | 0.00 | 0.32 | 0.00 | |
| CNMaR | FBgn0053696 | 0.00 | 0.00 | 0.00 | 0.00 | |
| CrzR | FBgn0036278 | 0.00 | 3.75 | 0.00 | 0.74 | |
| ETHR | FBgn0038874 | 0.00 | 0.66 | 0.00 | 0.00 | |
| FMRFaR | FBgn0035385 | 0.00 | 0.00 | 0.00 | 0.00 | |
| InR | FBgn0013984 | 5.98 | 8.11 | 0.00 | 7.22 | x |
| LpR1 | FBgn0066101 | 1.93 | 5.46 | 8.78 | 2.68 | |
| LpR2 | FBgn0051092 | 8.89 | 7.26 | 8.29 | 4.48 | x |
| PK1-R | FBgn0038201 | 0.00 | 0.00 | 0.00 | 0.00 | |
| Proc-R | FBgn0029723 | 0.00 | 0.00 | 0.32 | 5.39 | |
| rk* | FBgn0003255 | 0.00 | *2.49* | 0.00 | 0.00 | |
| RYa-R | FBgn0004842 | 0.00 | 0.00 | 0.00 | 0.29 | |
| SPR* | FBgn0029768 | *0.00* | *0.00* | *1.15* | *2.68* | |
| TakR86C | FBgn0004841 | 0.00 | 0.00 | 0.00 | 0.00 | |
| TkR99D | FBgn0004622 | 0.00 | 0.00 | 0.00 | 0.00 | |
| --Serotonin Receptors-- | | | | | | |
| 5-HT1A | FBgn0004168 | 0.00 | 0.37 | 0.00 | 0.00 | |
| 5-HT1B | FBgn0263116 | 0.00 | 4.20 | 0.00 | 1.37 | |
| 5-HT2A | FBgn0087012 | 0.33 | 6.15 | 0.00 | 7.37 | x |

*(Continued)*

**Table 2.** (Continued)

| Gene Name | Flybase Gene Number (FBgn) | Cell 1 | Cell 2 | Cell 3 | Cell 4 | Tested in RNAi Screen? |
|---|---|---|---|---|---|---|
| 5-HT2B | FBgn0261929 | 0.00 | 0.00 | 0.00 | 0.00 | |
| 5-HT7 | FBgn0004573 | 4.77 | 0.00 | 2.15 | 0.00 | |
| --Tyramine Receptors-- | | | | | | |
| TyR | FBgn0038542 | 0.00 | 0.00 | 0.00 | 0.00 | |
| TyRII | FBgn0038541 | 0.00 | 0.00 | 0.00 | 0.00 | |
| --Other-- | | | | | | |
| E75 | FBgn0000568 | 8.32 | 9.32 | 8.32 | 10.29 | x |
| E78 | FBgn0004865 | 0.00 | 2.88 | 0.00 | 3.19 | |
| EcR | FBgn0000546 | 6.99 | 6.88 | 0.00 | 8.11 | x |
| Egfr | FBgn0003731 | 3.07 | 6.51 | 0.32 | 4.82 | x |

Numbers are log2-transformed reads per million.

* indicates mapping ambiguity due to overlapping 3' region. Italics indicates a potential underestimation of actual read number due to this ambiguity. Shading indicates relative gene expression level, with hotter colors representing higher expression levels.

the genetic and molecular mechanisms through which these cells regulate circadian behavioral rhythms, we harvested individual non-DILP-expressing output cells from intact *Drosophila* brains (Fig 1A–1C), and performed scRNAseq to determine their transcriptional profile. In initial studies (reported in [4, 5]), we investigated the role of neuropeptides in rhythmic behaviors, and found two peptides whose expression in PI cells is necessary for robust rest:activity rhythms: DH44 and SIFa. Here, we further analyzed our sequencing results to identify other signaling molecules expressed by PI output cells that could underlie their ability to transmit circadian information.

Because PI cells lack molecular clocks, they must receive time-of-day information from core clock cells. We previously demonstrated that both DH44- and SIFa-expressing PI cells are anatomically connected to DN1 clock cells [5], however, it is unclear whether these connections constitute functionally significant synaptic inputs. Furthermore, it is unknown to what extent other clock cell populations provide inputs to PI cells, though it has long been appreciated that multiple groups of clock cells extend neuronal processes in close proximity to PI cell bodies and dendrites [37, 38]. Since many clock cells signal via release of neuropeptides, it is also possible that communication with PI cells could result from long-distance diffusion from processes not in close apposition to PI cells. To better understand potential clock cell regulation of PI output cells, we therefore mined our RNA sequencing dataset to look for expression of receptors for the major neuropeptides and small molecule neurotransmitters known to be released by central clock cells.

Central clock neurons use a variety of peptide neurotransmitters, including pigment dispersing factor (Pdf), which is expressed by sLNvs [39, 40], neuropeptide F (NPF), short neuropeptide F (sNPF) and ion transport peptide (ITP), which are expressed by subsets of LNvs and LNds [41], and DH31, Allatostatin-C and CCH1amide, which are expressed by subsets of DN1 cells [42–44]. We found limited evidence for expression of receptors for most of these peptides within PI output cells (Table 1). Notably, however, we did record substantial expression of the gene encoding for the Pdf receptor (*Pdfr*) in 3 out of 4 cells analyzed, demonstrating the potential for direct signaling between sLNv clock cells and PI output cells. We also found that 2 of 4 cells expressed the *sNPF receptor* gene at significant levels, providing additional support for the possibility of sLNv to PI cell communication. Interestingly, both PDF and sNPF were recently

**Table 3. Ion channel gene expression in PI output cells.**

| Gene Name | Flybase Gene Number (FBgn) | Cell 1 | Cell 2 | Cell 3 | Cell 4 | Tested in RNAi Screen? |
|---|---|---|---|---|---|---|
| | | --sodium channels-- | | | | |
| para | FBgn0264255 | 1.02 | 7.51 | 9.76 | 7.15 | |
| | | --potassium channels-- | | | | |
| eag | FBgn0000535 | 2.48 | 5.98 | 3.70 | 6.93 | x |
| elk | FBgn0011589 | 0.60 | 0.00 | 0.00 | 0.29 | |
| Hk | FBgn0263220 | 7.11 | 7.18 | 1.31 | 8.87 | x |
| Ih | FBgn0263397 | 8.01 | 9.49 | 6.87 | 9.75 | x |
| Irk1 | FBgn0265042 | 0.00 | 1.96 | 0.00 | 6.15 | |
| Irk2 | FBgn0039081 | 1.93 | 6.96 | 6.95 | 8.58 | x |
| Irk3* | FBgn0032706 | 0.00 | 3.14 | *0.00* | 0.00 | |
| KCNQ* | FBgn0033494 | 0.00 | 0.37 | 0.00 | *3.84* | |
| Ork1 | FBgn0017561 | 0.33 | 5.96 | 0.32 | 5.75 | |
| sand* | FBgn0033257 | 1.02 | *0.00* | 0.00 | 0.29 | |
| sei* | FBgn0003353 | 0.00 | 0.00 | 3.06 | 0.00 | |
| Shab | FBgn0262593 | 0.82 | 3.93 | 6.41 | 4.65 | |
| Shal | FBgn0005564 | 0.33 | 4.76 | 0.32 | 0.00 | |
| Shaw | FBgn0003386 | 0.82 | 0.37 | 2.07 | 1.80 | |
| Sh | FBgn0003380 | 2.83 | 7.35 | 9.50 | 10.45 | x |
| slo | FBgn0003429 | 5.43 | 7.13 | 9.36 | 7.70 | x |
| slo2 | FBgn0261698 | 0.33 | 6.09 | 0.00 | 3.97 | |
| SK | FBgn0029761 | 5.99 | 6.83 | 5.98 | 9.11 | x |
| 14-3-3zeta | FBgn0004907 | 9.83 | 11.10 | 10.15 | 11.65 | x |
| qvr (sss) | FBgn0260499 | 3.11 | 6.36 | 8.39 | 8.80 | x |
| slob | FBgn0264087 | 6.89 | 6.09 | 0.32 | 7.00 | x |
| | | --calcium channels-- | | | | |
| Ca-α1D | FBgn0001991 | 0.00 | 1.45 | 2.73 | 5.99 | |
| Ca-α1T | FBgn0264386 | 0.82 | 6.37 | 7.83 | 3.39 | |
| cac | FBgn0263111 | 1.61 | 5.04 | 6.75 | 7.55 | |
| Ca-β | FBgn0259822 | 0.60 | 0.90 | 3.06 | 6.65 | |
| CG4587 | FBgn0028863 | 0.33 | 4.52 | 6.51 | 5.47 | |
| stj | FBgn0261041 | 0.82 | 0.00 | 8.14 | 3.29 | |
| | | --chloride channels-- | | | | |
| ClC-a | FBgn0051116 | 4.67 | 4.15 | 0.00 | 5.91 | |
| ClC-b* | FBgn0033755 | 5.57 | 4.65 | 0.00 | 0.00 | |
| ClC-c* | FBgn0036566 | *0.00* | 3.48 | *5.22* | *5.52* | x |
| subdued | FBgn0038721 | 0.00 | 3.65 | 1.57 | 5.02 | |
| | | --cation channels-- | | | | |
| bib | FBgn0000180 | 0.00 | 3.28 | 0.00 | 0.00 | |
| na | FBgn0002917 | 0.33 | 0.00 | 2.78 | 0.00 | |

Numbers are log2-transformed reads per million.

* indicates mapping ambiguity due to overlapping 3' region. Italics indicates a potential underestimation of actual read number due to this ambiguity. Shading indicates relative gene expression level, with hotter colors representing higher expression levels.

shown to act on nearby DILP2-expressing PI cells, indicating that the sLNvs may directly communicate with multiple PI cell subsets [45].

In addition to peptides, core clock cells are also thought to release small molecule neurotransmitters, including glutamate (Glu) by DN1s [46] and acetylcholine (Ach) by LNds [34,

41]. In contrast to peptide receptors, we observed significant expression across multiple PI output cells of genes encoding for several Glu and Ach receptor subtypes (Table 1). Glutamate receptors are broadly divided into metabotropic and ionotropic types. *Drosophila* has a single functional metabotropic glutamate receptor (*DmGluRA*), which was expressed in 1 of 4 cells we analyzed. We also observed expression of the ionotropic *GluR1A*, *GluR1B*, *GluRIIE*, and *CG9935* subunits in 2 of 4 cells, and expression in 3 cells of *CG11155* and *GluCl*, the latter of which forms an inhibitory glutamate-gated chloride channel [47]. For ACh, we found significant expression in multiple PI cells of most of the nicotinic acetylcholine receptor (nAchR) α subunits, as well as a single β subunit (nAchRβ1). We found comparatively little expression of muscarinic AchRs. Taken together, these results indicate that PI cells possess the molecular substrates to receive circadian signals from both LNd and DN1 cells, via the small molecule neurotransmitters Glu and Ach.

We also determined expression of receptors for several additional neuropeptides that have been implicated in regulation of circadian rhythms, but which are not expressed by central clock cells. We observed low levels of expression in PI cells of the receptors for leucokinin (*Lkr*) [19], DH44 (*DH44-R1* and *DH44-R2*) [5], SIFa (*SIFaR*) [4], and hugin (*PK2-R1* and *PK2-R2*) [9] (Table 1), suggesting that these peptides do not strongly regulate PI cell function.

**PI cell expression of non-circadian receptors.**   PI output cells likely integrate other inputs in addition to those provided by circadian clock neurons, for example, signals involved in the communication of metabolic, mating, or sleep status [6]. We therefore analyzed our RNA sequencing results to determine expression of receptors for common neurotransmitters in the fly that are not thought to be released by neuronal populations that contribute to circadian circuits (Table 2). Multiple small molecule receptor types were highly and consistently expressed among analyzed cells, including the dopamine receptors *Dop2R* and *DopEcr*, the ionotropic GABA receptor *Rdl*, and the *Octβ2R* octopamine receptor. We additionally observed substantial expression of several neuropeptide receptors, including the insulin-like receptor (*InR*), the adipokinetic hormone receptor (*AkhR*), and the lipophorin receptors *LpR1* and *LpR2*.

**PI cell expression of ion channels that regulate cell excitability.**   Finally, we determined the expression of ion channels within PI output cells (Table 3), as these are important regulators of cell excitability and may contribute to the neuronal activity cycles that have been observed in PI cells [10, 18, 19]. Not surprisingly, we found evidence for expression of most major voltage-gated ion channels involved in action potential generation and propagation. We also observed expression of ion channels previously implicated in sleep and circadian rhythm regulation, including the *Shaker* (*sh*) [48] and *slo* [49, 50] potassium channels as well as associated subunits *Hyperkinetic* (*Hk*) [51], *sleepless* (*sss*) [52], and *slowpoke binding protein* (*Slob*) [53].

**A screen for circadian output genes.**   The expression of genes encoding for receptors and ion channels within PI output cells does not guarantee a role for those genes in regulating circadian outputs; it merely suggests the possibility of such regulation. Therefore, to test for a functional contribution to circadian outputs, we undertook a behavioral screen in which we measured rest:activity rhythm strength following PI-cell specific RNAi-mediated knockdown of candidate signaling molecules identified through our single-cell sequencing analysis. To restrict knockdown to relevant PI output cells, we used a combination of the SIFa-GAL4 and DH44-GAL4 lines to drive UAS-RNAi expression. We call this combined GAL4 line (which also includes UAS-Dicer2 to increase RNAi efficiency) SIFa/DH44-GAL4. In the brain, SIFa/DH44-GAL4 is restricted to 10 PI cells (Fig 2A), with additional sparse expression in a handful of presumptive neurons in the ventral nerve cord (not shown).

We screened 80 RNAi lines targeting a total of 38 genes (Fig 2B; S2 File). In most cases, this included multiple RNAi lines targeting a given gene. For 12 of these lines (targeting 10 unique

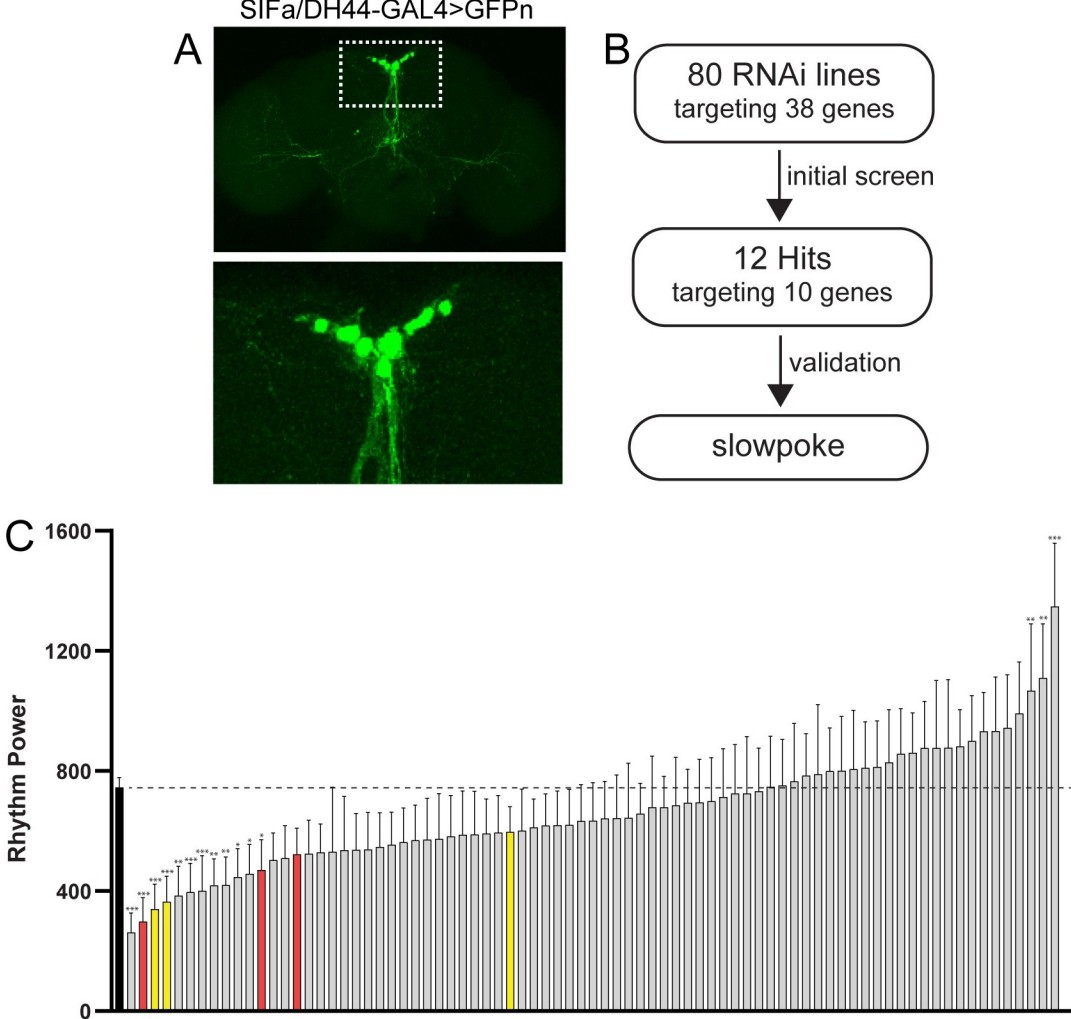

**Fig 2. A screen for circadian output genes that function in PI cells.** (A) Representative maximum projection confocal images of a fly brain in which a combination of SIFa-GAL4 and DH44-GAL4 lines were used to drive expression of a nuclear-localized GFP (SIFa/DH44-GAL4>GFPn). GFP expression in the brain is limited to 10 cells in the PI. The PI region (delineated by the dashed rectangle on the top image) is shown in a magnified view below. (B) Experimental design of our behavioral screen. We used the SIFa/DH44-GAL4 line to drive expression of 80 different RNA lines (targeting a total of 38 genes) specifically in PI output cells. Through screening and validation, we identified the *slo* potassium channel as a circadian output gene in the PI. (C) Screen results depict rest:activity rhythm power (mean ± 95% confidence interval) for all 80 experimental lines (in which SIFa/DH44-GAL4 was used to drive UAS-RNAi expression) as well as for GAL4 control flies (black bar). *p <0.01, **p < 0.001, ***p < 0.0001 compared to GAL4 control flies, Dunnett's multiple comparisons test. Genes for which multiple independent lines targeting the same gene resulted in reduced rest:activity rhythm strength are labeled in red (*slo*) and yellow (*sss*). See S2 File for detailed information on rest:activity rhythm power, period and *n* for each line.

genes), knockdown resulted in a significant reduction in rhythm strength compared to control SIFa/DH44-GAL4 flies (Fig 2C). These "hit" lines targeted an array of genes, including both receptors and ion channels. Among receptors were those encoding for 3 nicotinic AchR α subunits (*nAChRα1*, *nAChRα3*, and *nAChRα6*), 2 ionotropic glutamate receptor subunits (*CG9935*/*Ekar* and *CG11155*), and the *5-HT2A* serotonin receptor. We additionally observed decreased rest:activity rhythm strength associated with PI-specific knockdown of the potassium channel genes *sss* and *slo*, as well as 14-3-3ζ, which regulates *slo* activity [54]. In contrast to these changes in rhythm strength, we found little evidence for alteration of period length

associated with PI-selective knockdown of any of our candidate genes (S1 Fig; S2 File). This is consistent with the idea that PI cells regulate circadian outputs, rather than directly affecting the core pacemaker.

Given the potential for false positives associated with large screens, we conducted several additional analyses to confirm a role for these genes in the circadian output function of SIFa- and DH44-expressing PI cells. First, we assessed for effects on rest:activity rhythms of gene knockdown using the same UAS-RNAi lines, but instead drove expression in the IPCs of the PI with a highly specific DILP2-GAL4 line (Fig 3A). As the IPCs have not been implicated in regulation of behavioral rhythms, we reasoned that knockdown in these cells should not affect rest:activity rhythm strength, thus allowing us to identify any potential non-specific effects on rhythmicity. We observed reduced rest:activity rhythm strength in 2 of 12 "hit" lines when tested with DILP2-GAL4: one targeting *CG9935*, and another targeting *nAChRα1* (Fig 3B). For the remaining 10 lines, we saw no effect of IPC-specific knockdown on rest:activity rhythms. We interpret these results as indicative of non-specific effects associated with the *CG9935* and *nAChRα1*-targeting lines (though we cannot rule out an indirect consequence of IPC manipulation on rest:activity rhythm strength). In contrast, the lack of effect in the remaining 10 lines supports an output function of these genes and furthermore demonstrates cellular specificity of action, especially because the IPCs constitute a nearby population of cells in the same brain region as those expressing SIFa and DH44.

Second, to address the issue of possible off-target effects associated with RNAi, we looked for evidence of consistent effects across multiple, independently generated RNAi lines targeting different regions of a gene. Somewhat surprisingly, we observed consistent behavioral effects for only 2 of the 10 genes identified in our screen: *slo* and *sss*. For the remaining 8 genes, we failed to replicate our findings with additional RNAi lines targeting the same genes. Thus, we cannot rule out the possibility that the reduced rest:activity rhythm strength in these 8 lines is due to off-target RNAi effects. In contrast, the similar phenotypes observed in multiple RNAi lines targeting *slo* and *sss* argue against off-target effects as accounting for the observed phenotype, thereby providing support for a role for these genes in regulating rest: activity rhythm outputs.

Importantly, for these two genes, we confirmed the findings of our initial screen in a set of follow-up experiments with increased sample size. For *slo*, we found a significant reduction in rest:activity rhythm strength for 2 of 3 RNAi lines (with a non-significant trend towards reduction with the third line) when driven by SIFa/DH44-GAL4 (Fig 4A). In general, these flies retained some residual rhythmicity following PI selective *slo* knockdown, but activity patterns were messier, with more activity occurring during times of normal quiescence as compared to controls (Fig 4B). Although reduction in rest:activity rhythm strength tended to be subtler following *sss* knockdown in PI output cells as compared to *slo*, we recorded significant effects in all 3 *sss*-targeting RNAi lines (Fig 4C). As was the case with *slo* knockdown, most of these flies retained some semblance of rhythmicity, but with less consolidated periods of rest and activity (Fig 4D).

As a final test of specificity, we drove expression of *sss* and *slo* RNAi constructs with C929-GAL4, which labels the vast majority of PI cells (Fig 5A), including those expressing SIFa and DH44 peptides [55]. Outside of the PI, C929-GAL4 is also expressed in a number of other peptide-expressing cells in the brain. Given the common expression in the PI cells of interest, we reasoned that knockdown using this GAL4 line should recapitulate findings with SIFa/DH44-GAL4. Interestingly, we observed divergent effects of *slo* and *sss* knockdown using C929-GAL4. Whereas C929-GAL4-mediated expression of all 3 *slo* targeting RNAi constructs significantly decreased rest:activity rhythm strength (Fig 5B and 5C), none of the *sss*-targeting lines produced a significant effect (Fig 5D). We note, however, that even in the case of *slo*

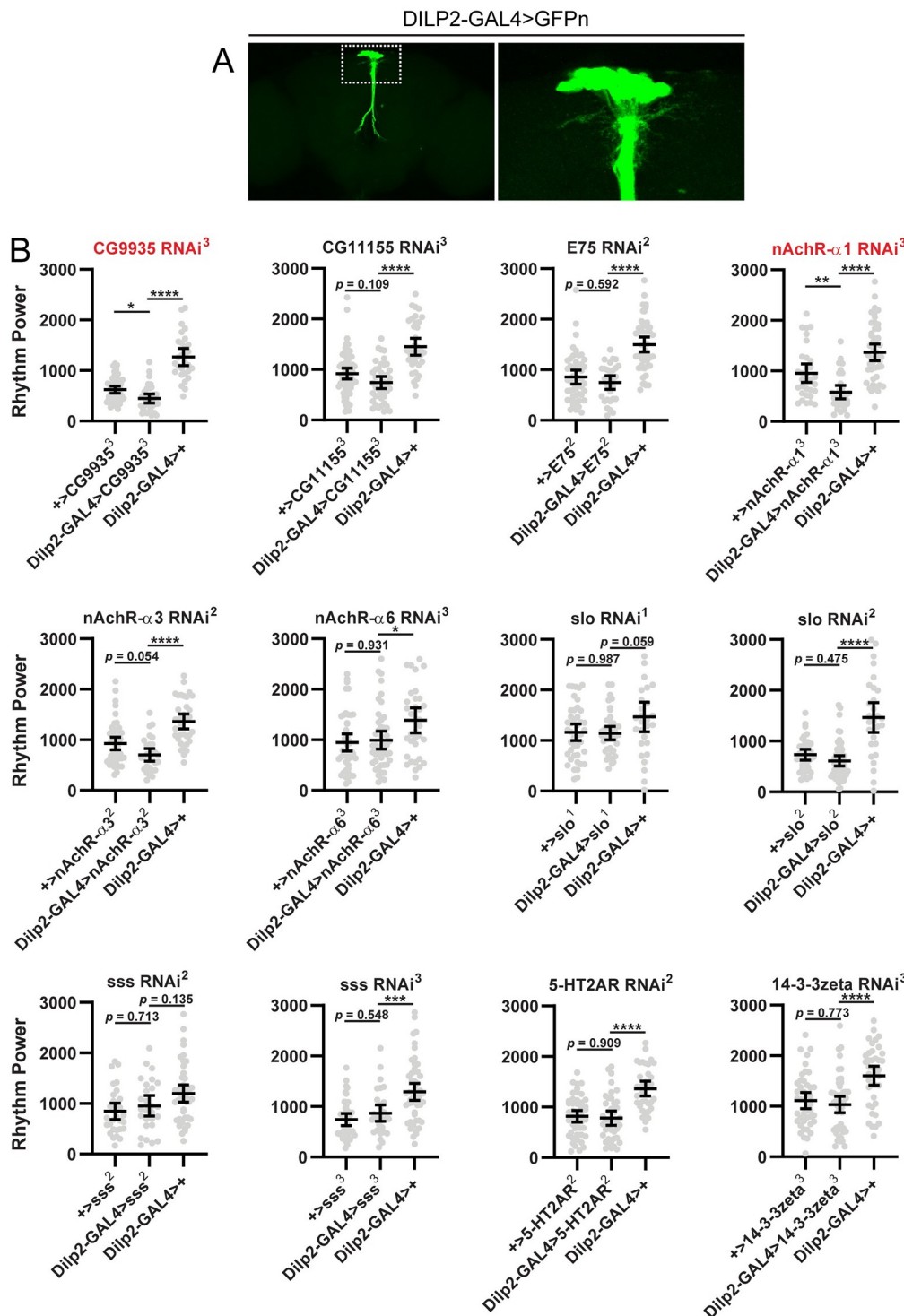

**Fig 3. Effect of RNAi-mediated knockdown of candidate output genes in IPCs.** (A) Representative maximum projection confocal images of a fly brain in which DILP2-GAL4 was used to drive expression of a nuclear-localized GFP (DILP2-GAL4>GFPn). GFP expression is limited to ~14 cells in the PI region. The right panel shows a magnified view of the boxed region on the left panel. (B) Rest:activity rhythm power is displayed for the genotypes listed. Lines are means ± 95% confidence intervals. Dots represent individual flies. $^*p < 0.05$, $^{**}p < 0.01$, $^{***}p < 0.001$, $^{****}p < 0.0001$, Tukey's multiple comparisons test for experimental line compared to GAL4 or UAS controls. To simplify nomenclature, DILP2-GAL4>xxx refers to flies in which DILP2-GAL4 has been used to drive expression of an RNAi construct targeting gene xxx. + represents a wildtype chromosome. Graphs labeled in red indicate experiments for

which we observed significantly reduced rest:activity rhythm strength in experimental flies compared to both GAL4 and UAS controls. Only *CG9935*- and *nAchRα1*- targeting RNAi constructs produced significant effects compared to both controls. See S2 File for detailed information on rest:activity rhythm power, period and *n* for each line.

knockdown, the phenotype was milder than that observed with the SIFa/DH44-GAL4 driver (Fig 5B and 5C). Thus it is possibility that SIFa/DH44-GAL4 drives more robust expression than C929-GAL4, accounting for the lack of effect of *sss* manipulations using the latter driver. Conservatively, however, we conclude that, among all genes tested, *slo* is the only gene for which we have unequivocally demonstrated a role as a circadian output gene within PI output cells.

## Discussion

Core clock neurons in the brain modulate behavior through circadian output circuits that ultimately connect the clock cells to downstream neuronal populations that control overt behaviors. Output pathways are among the least well understood aspects of circadian rhythm regulation. To better characterize output circuits governing the generation of circadian rest: activity rhythms, we took a twofold approach. First, we used scRNAseq to identify potential circadian output genes expressed by cells in the PI region of the fly brain. We focused on genes with known roles in neuronal communication and excitability with the idea that such genes would be involved in receiving circadian information from clock cells and transmitting it to downstream components of the output circuit. Second, we assessed the behavioral consequences of RNAi-mediated knockdown of these genes within PI output cells. Because it is likely that many output molecules are essential to other important cell functions, global elimination of these genes, such as occurs in mutant lines, may result in pleiotropic effects or even developmental lethality. Our strategy therefore offers benefits compared to traditional approaches used to identify gene function in *Drosophila*, such as forward genetic screens, because it allows us to assess a specific function of these genes in PI neurons.

We hypothesized that this approach would inform our understanding of the manner through which circadian information is transmitted out of the clock cell network to downstream output cells. For example, it is currently unknown whether distinct clock cell groups give rise to multiple parallel downstream pathways or whether circadian information is first consolidated in select clock cell populations before being transmitted to output cells. By targeting receptors in PI cells of neurotransmitters and neuropeptides known to be used by specific clock cells, we reasoned that we would be able to pinpoint clock cell populations whose input to the PI is necessary to maintain robust circadian rhythms. Importantly, because PI manipulations do not alter clock cell function [5], such an approach should allow for isolation of the contribution of specific clock cells to behavioral and physiological outputs without disrupting the overall function of the clock cell network. Surprisingly, however, though we observed PI cell expression of genes encoding for multiple neurotransmitters and neuropeptides expressed by clock cells, we found no consistent effect of RNAi-mediated downregulation of those receptors on rest:activity rhythms. This lack of effect could result from molecular redundancy, especially in the case of the neurotransmitters glutamate and acetylcholine, for which multiple receptor subtypes were expressed by the same PI cells. This issue could be circumvented by simultaneously targeting multiple receptor subtypes, although this is an experimentally difficult undertaking.

One limitation of this study was that our scRNAseq analysis was conducted on a relatively small number of PI cells. Increasing the sample size would help to better understand the potential heterogeneity of gene expression between cells. When determining targets for our RNAi

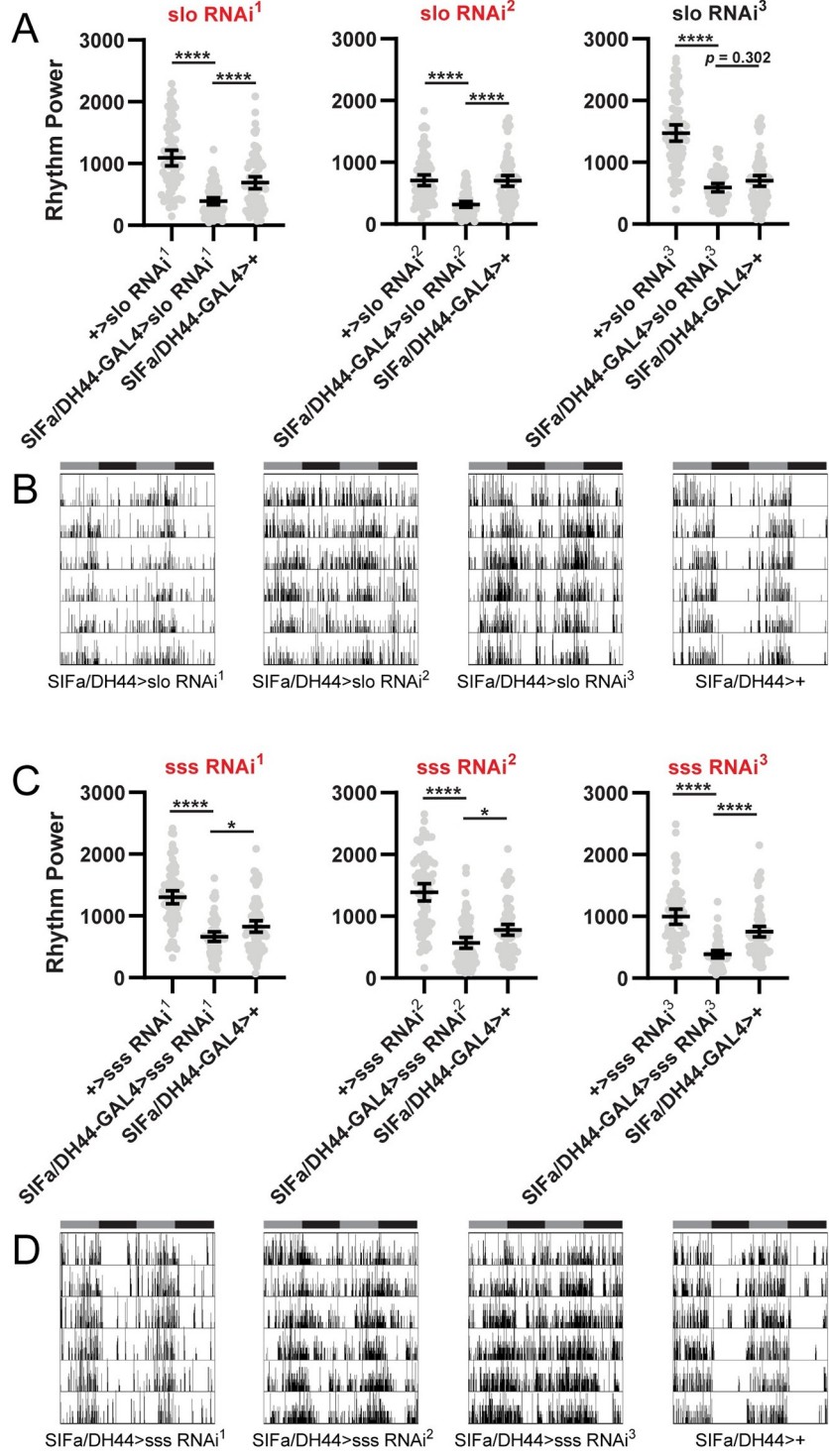

**Fig 4. PI-specific knockdown of *slo* or *sss* reduces rest:activity rhythm strength.** (A) Rest:activity rhythm power is displayed for flies in which SIFa/DH44-GAL4 was used to drive expression of 3 independent RNAi constructs targeting the *slo* gene. Lines are means ± 95% confidence intervals. Dots represent individual flies. *$p < 0.05$, **$p < 0.01$, ***$p < 0.001$, ****$p < 0.0001$, Tukey's multiple comparisons test for experimental line compared to GAL4 or UAS controls. (B) Representative single fly activity records over 6 days in DD for the genotypes listed. Activity in infrared beam breaks/min is plotted for each minute. Activity records are double plotted, with 48 hours of data on each line and the second 24 hours replotted at the start of the next line. Gray and black bars above each plot represent subjective day and night, respectively. (C) Rest:activity rhythm power is displayed as described in (A) for flies in which

SIFa/DH44-GAL4 was used to drive 3 independent RNAi constructs targeting the *sss* gene. (D) Representative single fly activity records over 6 days in DD are displayed as described in (B) for the genotypes listed. Graphs labeled in red indicate experiments for which we observed significantly reduced rest:activity rhythm strength in experimental flies compared to both GAL4 and UAS controls. We noted significant reduction in rest:activity rhythm strength following SIFa/DH44-GAL4-mediated expression of 2 of 3 *slo*-targeting RNAi constructs, and 3 of 3 *sss*-targeting constructs. See S2 File for detailed information on rest:activity rhythm power, period and *n* for each line.

screen, we sought out signaling genes with significant expression in multiple analyzed cells because we thought that such genes would be more likely to play an important output role. However, it is possible that we have missed relevant genes that are expressed heterogeneously within the PI. For example, several glutamate receptor subunits were expressed by only 1–2 cells out of the 4 that we analyzed, and these were not tested for a functional role in circadian rhythm regulation.

We also note that sequencing analysis revealed expression of genes that are thought to selectively label non-neuronal cells, including glial and fat body cells. We also observed expression of a number of genes typically associated with eye photoreceptors, including several rhodopsin genes and the glutamate receptor subunit *CG9935/Ekar* [56]. This unexpected gene expression could result from contamination, for example if debris from non-PI cells entered the pipet during cell harvesting, and we cannot rule out this possibility. Because of this, we caution that gene expression patterns suggested by our scRNAseq analysis should be independently confirmed.

Despite these limitations, our approach successfully uncovered an essential function of the *slo* gene in regulating the circadian output function of SIFa- and DH44-expressing PI neurons. *Slo* knockdown with multiple distinct RNAi constructs significantly attenuated rest:activity rhythm strength. Furthermore, this effect was cell specific, as expression of the same RNAi constructs in nearby IPCs had no impact on rest:activity rhythms. *Slo* is a member of the "Big K" family of voltage-gated calcium-dependent potassium channels [57, 58], which regulate cell excitability in part through effects on repolarization following action potentials. Interestingly, previous studies suggested a role for *slo* in the generation of rest:activity rhythms, as neuron-specific *slo* mutants are largely arrhythmic [49, 50]. "Big K" potassium channels perform a similar function in mammals, and mutations in *Kcnma1*, which encodes for a mammalian "Big K" channel, degrade rest:activity rhythms in mice [59].

*Slo* functions in part as an output molecule within circadian neurons. Although molecular clock cycling is intact in sLNv clock cells of *slo* mutants, clocks in dorsal clock neurons become desynchronized, suggesting a role for *slo* in communication between sLNvs and other parts of the clock network. However, rescue of *slo* in all clock cells does not fully reestablish behavioral rhythms, which indicates that expression in non-clock neurons is also necessary [49]. In conjunction with these previous results, our findings demonstrate that *slo* exerts it effects in multiple components of the circadian circuit, including PI output cells in addition to clock neurons.

Interestingly, such an arrangement, in which an output molecule acts in multiple nodes of the circadian output circuit, has also been proposed for other previously identified output molecules, including neurofibromin, the protein product of the disease-related *Neurofibromatosis 1* gene [18], Dyschronic, which regulates Slowpoke expression [60], and the RNA binding protein, LARK [61]. It is unclear whether these genes play a specific role in regulating circadian-relevant neurons, or whether they underlie more general aspects of neuronal physiology, such that their loss impacts any functions subserved by the neurons in which knockdown or mutation occurs. In the case of *slo*, the conserved circadian function in flies and mammals argues for an important and specific contribution to circadian rhythm regulation, likely by

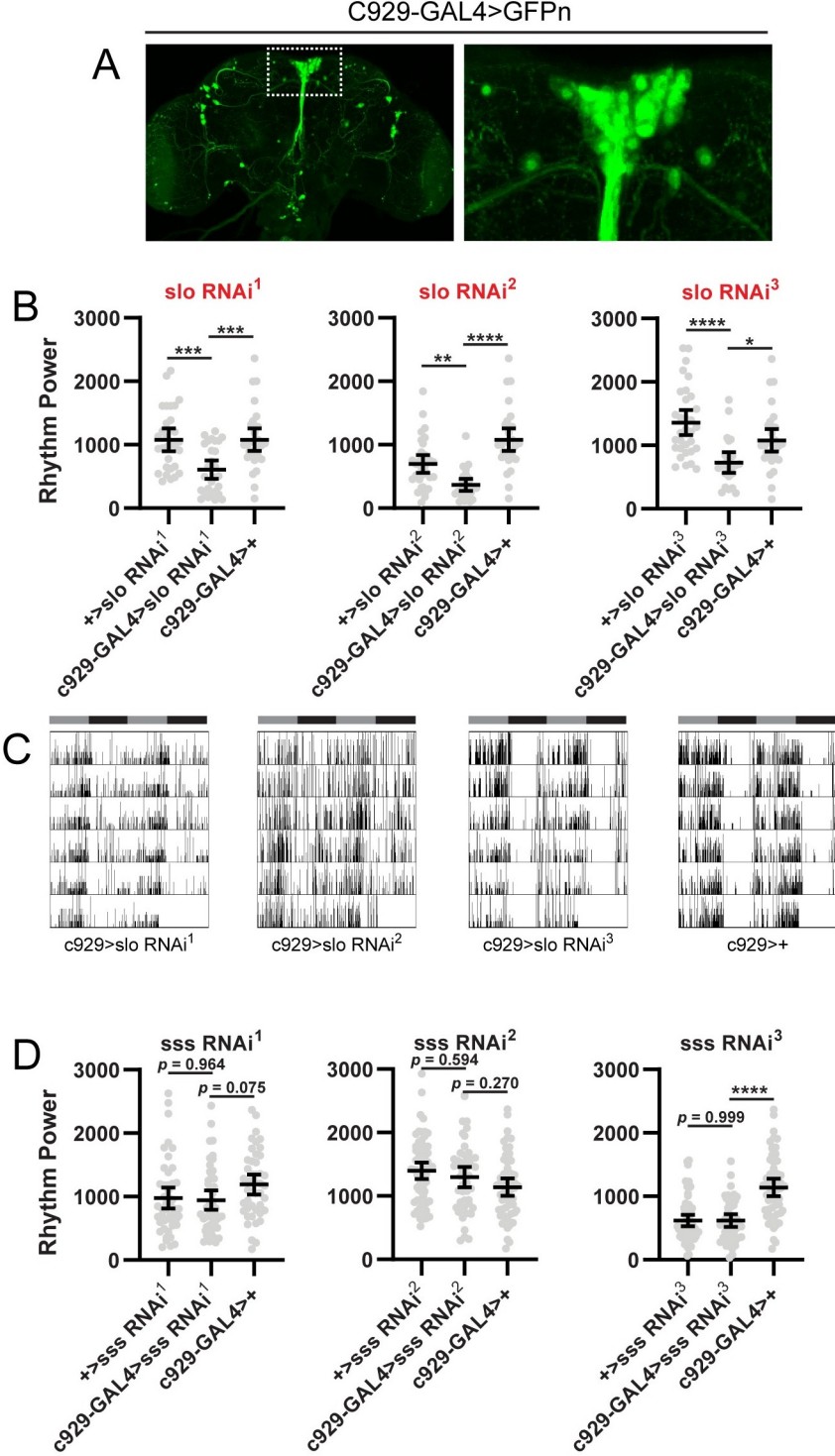

**Fig 5. Confirmation of *slo* as a circadian output gene.** (A) Representative maximum projection confocal images of a fly brain in which C929-GAL4 was used to drive expression of a nuclear-localized GFP (C929-GAL4>GFPn). GFP is expressed in all PI cells plus a number of peptidergic neurons outside the PI. The right panel shows a magnified view of the boxed region on the left panel. (B) Rest:activity rhythm power is displayed for flies in which C929-GAL4 was used to drive expression of 3 independent RNAi constructs targeting the *slo* gene. Lines are means ± 95% confidence intervals. Dots represent individual flies. *$p < 0.05$, **$p < 0.01$, ***$p < 0.001$, ****$p < 0.0001$, Tukey's multiple comparisons test for experimental line compared to GAL4 or UAS controls. (C) Representative single fly activity

records over 6 days in DD for the genotypes listed. (D) Rest:activity rhythm power is displayed as described in (A) for flies in which C929-GAL4 was used to drive 3 independent RNAi constructs targeting the *sss* gene. Graphs labeled in red indicate experiments for which we observed significantly reduced rest:activity rhythm strength in experimental flies compared to both GAL4 and UAS controls. C929-GAL4-mediated expression of all 3 *slo*-targeting RNAi constructs significantly reduced rest:activity rhythm strength, but expression of *sss*-targeting RNAi constructs was without effect. See S2 File for detailed information on rest:activity rhythm power, period and *n* for each line.

contributing to rhythmic neuronal excitability that allows for circadian information to propagate across output circuits.

## Supporting information

**S1 File. List of RNAi lines used in behavioral screening.**
(XLSX)

**S2 File. Rest:activity rhythm power and period data for all experimental manipulations.**
(XLSX)

**S1 Fig. Lack of effect of PI-specific knockdown of candidate circadian output genes on rest:activity rhythm period.** Screen results depict rest:activity rhythm period (mean ± 95% confidence interval) for all 80 experimental lines (in which SIFa/DH44-GAL4 was used to drive UAS-RNAi expression) as well as for GAL4 control flies (black bar). Only two lines—nAchRα3 RNAi[1] (red bar) and EcR RNAi[1] (yellow bar)—exhibited a statistically significant difference in period compared to control flies, and even in these cases, the effect sizes were small and inconsistent across other RNAi lines targeting these same genes. *p $<$0.05, ***p $<$ 0.0001 compared to GAL4 control flies, Dunnett's multiple comparisons test.
(TIF)

## Acknowledgments

We thank Drs. Julian Wooltorton, Ana Lia Obaid, Jennifer Spaethling, James Eberwine and Amita Sehgal for assistance with PI cell harvesting, imaging, RNA amplification and library prep.

## Author Contributions

**Conceptualization:** Daniel J. Cavanaugh.

**Data curation:** Daniela Ruiz, Saffia T. Bajwa, Naisarg Vanani, Tanvir A. Bajwa, Daniel J. Cavanaugh.

**Formal analysis:** Daniela Ruiz, Saffia T. Bajwa, Naisarg Vanani, Tanvir A. Bajwa, Daniel J. Cavanaugh.

**Funding acquisition:** Daniel J. Cavanaugh.

**Investigation:** Daniela Ruiz, Saffia T. Bajwa, Naisarg Vanani, Tanvir A. Bajwa, Daniel J. Cavanaugh.

**Methodology:** Daniel J. Cavanaugh.

**Project administration:** Daniel J. Cavanaugh.

**Supervision:** Daniel J. Cavanaugh.

**Writing – original draft:** Daniel J. Cavanaugh.

**Writing – review & editing:** Daniela Ruiz, Saffia T. Bajwa, Naisarg Vanani, Tanvir A. Bajwa, Daniel J. Cavanaugh.

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
