## [Decision Letter · Decision Letter 0]

1 Feb 2021

PONE-D-20-37778

Slowpoke functions in circadian output cells to regulate rest:activity rhythms

PLOS ONE

Dear Dr. Cavanaugh,

Thank you for submitting your manuscript to PLOS ONE. After careful consideration, we feel that it has merit but does not fully meet PLOS ONE’s publication criteria as it currently stands. Therefore, we invite you to submit a revised version of the manuscript that addresses the points raised during the review process.

Both reviewers have found this manuscript very interesting and worthy of publication, however, only after addressing a number of issues that the authors should address in full in their revised version.

I would in particular advise the authors to focus on the following points:

The inclusion of a scheme that could help the non-specialist reader to follow the location and neurotransmitter expression profile of the PI cells under investigation.

Clarification of how the cells for the RNAseq analysis were isolated.

Improving the statistical analysis of the "power" of the circadian rhythmicity

We look forward to receiving your revised manuscript.

Kind regards,

Nicholas Simon Foulkes, D.Phil

Academic Editor

PLOS ONE

Journal Requirements:

Reviewers' comments:

Reviewer's Responses to Questions

**Comments to the Author**

1. Is the manuscript technically sound, and do the data support the conclusions?

Reviewer #1: Partly

Reviewer #2: Yes

2. Has the statistical analysis been performed appropriately and rigorously? 

Reviewer #1: Yes

Reviewer #2: I Don't Know

3. Have the authors made all data underlying the findings in their manuscript fully available?

Reviewer #1: Yes

Reviewer #2: Yes

4. Is the manuscript presented in an intelligible fashion and written in standard English?

Reviewer #1: Yes

Reviewer #2: Yes

5. Review Comments to the Author

Reviewer #1: I found this to be an interesting and very well written ms, which after the single cell sequencing. was quite disappointing in terms of assigning function to the transcripts identified. here are some comments

In the Methods, it would be useful just to state briefly where each of the gal4 drivers is expressed. For example why would the 5 cells labelled by kurs58-GAL4 or C767-GAL4 be of particular interest other than they are in the PI? What exactly is dilp2mCherry? I do not want to trawl through other papers to find out. Could the expression patterns and a description of the lines be added to Table S1? A non-Drosophilist would really struggle with what exactly was done and which cells were labelled. Perhaps a cartoon of PI cells might be helpful indicating where the various subsets of cells expressing the various neuropeptides are located?

L239 and l256-7 I seem to recall a paper by Nagy et al from Costa’s group a few years ago that showed that PDF clock cells are also connect to dilp2 cells??? Might this also be relevant here?

I appreciate that the authors wish to look at the functional effects of gene knockdown on output so have limited their analysis to the ‘power’ of rhythms. They assume that because the core clock is not affected the free-running periods will be ~24 h. I’m wondering why they did not examine the periods though, because there is a possibility that there is feedback between output cells and the clock – or even off target effects. The period data must be there, why not show it as there could be something interesting. I’m a little concerned about the measure of power too. Chi-2 periodograms are quite crude compared to more recent methods which also generate power values. Even a simple autocorrelation would generate a more robust power value. However, I realise that a lot of studies do use this measure of power so I won’t insist on a different measure.

The authors are quite honest about the variability of their results, for which they should be commended. Apart from slo, no clear pattern emerges about the relevance of the other genes at a functional level. This might be perhaps because the authors only focused on power of circadian activity cycles. There could be effects on phase or period changes in activity or on eclosion rhythms, or in sleep – easy to measure but the authors did not explore these other phenotypes.

In conclusion, this is an interesting ms that at least shows what mRNAs are expressed in selected PI neurons. The functional tests reveal disappointing results. Nevertheless I think the ms is worth publishing.

Reviewer #2: The paper by Ruiz et al describes an interesting work that aims at isolating genes involved in previously defined output neurons of the drosophila brain circadian clock. A single cell RNAseq experiment with 4 cells of the pars intercerebralis generates a series of expressed genes encoding various neurotransmission components. Using targeted RNAi, the authors test the contribution of these components to the locomotor activity rhythms. They reveal that the slowpoke potassium channel plays a role in the PI cells to generate robust activity rhythms in constant conditions. Slowpoke was already known to be involved in the control of the circadian behavior but only clock cells were reported to be a site for slowpoke clock function, and the present study indicates that at least part of the non-clock cell function takes place in the PI.

The molecular and behavioral data are clearly presented and provide interesting information about slowpoke role in the clock neuron downstream circuit, which remains poorly understood.

My only comment is about the very limited description of the cell isolation procedure. I think that the authors should provide more information on how they isolate single cells for RNAseq analysis.

6. PLOS authors have the option to publish the peer review history of their article (what does this mean?). If published, this will include your full peer review and any attached files.

Reviewer #1: No

Reviewer #2: No

---

## [Author Response · Author response to Decision Letter 0]

14 Feb 2021

We thank the reviewers for their careful reading of our manuscript. Based on the reviewer feedback we have undertaken an extensive revision of the manuscript, and we believe the updated version to be significantly strengthened. This includes 1) adding needed detail regarding our rationale and scheme for identifying specific PI cell types for single-cell sequencing, 2) clarifying our single-cell harvesting protocol, 3) adding analysis of circadian period to supplement our power analysis (the results of which are now discussed in the text and included in S2 File and in a new supporting figure, S1 Figure).

We have outlined our responses below for each reviewer, including line references where appropriate, and feel the paper has been made substantially clearer and more complete as a result. We hope you agree it is now ready for publication.

Response to Academic Reviewer:

I would in particular advise the authors to focus on the following points:

The inclusion of a scheme that could help the non-specialist reader to follow the location and neurotransmitter expression profile of the PI cells under investigation.

We have now added extensive explanation/description of the expression profile of the PI cells under investigation (see response to Reviewer #1).

Clarification of how the cells for the RNAseq analysis were isolated.

We have now added clarification and detail about the protocol used to isolate single PI cells for RNAseq analysis (see response to Reviewer #2).

Improving the statistical analysis of the "power" of the circadian rhythmicity

As described in our response to Reviewer #1, we believe that our use of Chi-2 periodogram to assess “power” of rest:activity rhythms is appropriate, as evidenced by the fact that this method is by far the most commonly used in the field. We have provided documentation citing the use of Chi-2 periodogram in recent papers by many prominent labs. We appreciate the reviewer’s raising this concern; however, we feel it is best to retain use of the Chi-2 periodogram to facilitate comparisons with the vast majority of recent published work, in adherence with the general consensus among Drosophila chronobiology researchers regarding the suitability of the method.

Response to Reviewer #1:

Reviewer #1: I found this to be an interesting and very well written ms, which after the single cell sequencing. was quite disappointing in terms of assigning function to the transcripts identified. here are some comments

In the Methods, it would be useful just to state briefly where each of the gal4 drivers is expressed. For example why would the 5 cells labelled by kurs58-GAL4 or C767-GAL4 be of particular interest other than they are in the PI? What exactly is dilp2mCherry? I do not want to trawl through other papers to find out. Could the expression patterns and a description of the lines be added to Table S1? A non-Drosophilist would really struggle with what exactly was done and which cells were labelled. Perhaps a cartoon of PI cells might be helpful indicating where the various subsets of cells expressing the various neuropeptides are located?

We apologize for the lack of clarity. We have now added substantial wording to the methods section to better explain our rationale for capturing cells labelled by either kurs58-GAL4 or C767-GAL4 and also to explain the dilp2mCherry construct.

“We used a single-cell transcriptional profiling approach to identify potential circadian output genes expressed by relevant PI cell populations. The PI is comprised of ~30 cells, but only specific subsets have been implicated in control of rest:activity rhythms. Because the 14 DILP-expressing PI cells do not appear to contribute to rest:activity regulation (4,5), we sought to target non-DILP-expressing PI cells for single-cell sequencing following GFP-guided cell capture. To identify the cells of interest, we drove GFP expression with either of two GAL4 lines, kurs58-GAL4 or C767-GAL4, which are both active in non-DILP-expressing PI cells (5). Notably, constitutive neuronal activation under the control of either kurs58-GAL4 or C767-GAL4 compromises rest:activity rhythm strength, confirming the relevance of these cells (5). The flies used for single-cell capture also included a Dilp2mCherry construct, which selectively labels the DILP-expressing PI cells. This served two purposes: first, Dilp2mCherry acted as a landmark to aid in PI localization; second, it allowed us avoid selecting DILP-expressing cells, which could be easily identified based on their mCherry fluorescence (see Fig 1A-B).” (Lines 119-131).

L239 and l256-7 I seem to recall a paper by Nagy et al from Costa’s group a few years ago that showed that PDF clock cells are also connect to dilp2 cells??? Might this also be relevant here?

We thank the reviewer for bringing this paper to our attention. It is definitely relevant, and we have added a sentence indicating that the Nagy et al paper provides evidence for sLNv action on PI cells, consistent with our finding of the Pdf and sNPF receptor expression in our single-cell analysis:

“Interestingly, both PDF and sNPF were recently shown to act on nearby DILP2-expressing PI cells, indicating that the sLNvs may directly communicate with multiple PI cell subsets (45).” (Lines 286-288).

I appreciate that the authors wish to look at the functional effects of gene knockdown on output so have limited their analysis to the ‘power’ of rhythms. They assume that because the core clock is not affected the free-running periods will be ~24 h. I’m wondering why they did not examine the periods though, because there is a possibility that there is feedback between output cells and the clock – or even off target effects. The period data must be there, why not show it as there could be something interesting. 

We have now added data about period, in addition to our initial power analysis. We do not find evidence for period effects. In our initial screen, period length for all lines fell within a very small range. 2/80 lines tested did exhibit a statistically significant difference, but the magnitude of the effect was very small, and this was not consistent among other RNAi constructs targeting the same genes. We have added the period data from all of our experiments to S2 File, and additionally include a supporting figure (S1 Figure) depicting the period data from our initial screen. We have included a short discussion of these findings in the main text as well:

“In contrast to these changes in rhythm strength, we found little evidence for alteration of period length associated with PI-selective knockdown of any of our candidate genes (S1 Fig; S2 File). This is consistent with the idea that PI cells regulate circadian outputs, rather than directly affecting the core pacemaker.” (Lines 373-377).

I’m a little concerned about the measure of power too. Chi-2 periodograms are quite crude compared to more recent methods which also generate power values. Even a simple autocorrelation would generate a more robust power value. However, I realise that a lot of studies do use this measure of power so I won’t insist on a different measure.

We understand the reviewer’s concern with our choice of statistical method to measure circadian power. There are quite a few different types of tests that have been proposed to be useful in measuring circadian period and robustness, and none is perfect. However, as the reviewer notes, Chi-2 periodogram is used by many in the field. It is by far the most commonly used method to measure Drosophila rest:activity rhythms (for evidence of this, below we have included recent citations from many prominent labs in the field that have all used Chi-2 periodogram). We therefore suggest that Chi-2 periodogram is the most appropriate choice in our case, in order to maintain consistency with what others are doing and to allow for more direct comparisons between our work and that from other labs.

Allada Lab: Kula-Eversole et al (2021) Phosphatase of Regenerating Liver-1 Selectively Times Circadian Behavior in Darkness via Function in PDF Neurons and Dephosphorylation of TIMELESS. Curr Biol, 31(1):138-149.

Ceriani Lab: Herrero et al (2020) Coupling Neuropeptide Levels to Structural Plasticity in Drosophila Clock Neurons. Curr Biol, 30(16):3154-3166.

Hardin Lab: Gunawardhana et al (2017) VRILLE Controls PDF Neuropeptide Accumulation and Arborization Rhythms in Small Ventrolateral Neurons to Drive Rhythmic Behavior in Drosophila. Curr Biol, 27(22):3442-3453.

Rosbash Lab: Schlichting et al (2019) Neuron-specific knockouts indicate the importance of network communication to Drosophila rhythmicity. eLIFE, 8:e48301.

Rouyer Lab: Chatterjee et al (2019) Reconfiguration of a Multi-oscillator Network by Light in the Drosophila Circadian Clock. Curr Biol, 28(13):2007-2017.

Shafer Lab: Fernandez et al (2020) Sites of Circadian Clock Neuron Plasticity Mediate Sensory Integration and Entrainment. Curr Biol, 30(12):2225-2237.

Taghert Lab: Liang et al (2019) Morning and Evening Circadian Pacemakers Independently Drive Premotor Centers via a Specific Dopamine Relay. Neuron, 102(4):843-857.

The authors are quite honest about the variability of their results, for which they should be commended. Apart from slo, no clear pattern emerges about the relevance of the other genes at a functional level. This might be perhaps because the authors only focused on power of circadian activity cycles. There could be effects on phase or period changes in activity or on eclosion rhythms, or in sleep – easy to measure but the authors did not explore these other phenotypes.

In conclusion, this is an interesting ms that at least shows what mRNAs are expressed in selected PI neurons. The functional tests reveal disappointing results. Nevertheless I think the ms is worth publishing.

We appreciate that the reviewer sees the value in publication of our manuscript. While we have not focused here on other circadian outputs other than rest:activity rhythm strength (and now period), we agree that it would be of interest in future studies to assess the impact of our manipulations on other outcomes such as sleep and eclosion rhythms.

Response to Reviewer #2:

Reviewer #2: The paper by Ruiz et al describes an interesting work that aims at isolating genes involved in previously defined output neurons of the drosophila brain circadian clock. A single cell RNAseq experiment with 4 cells of the pars intercerebralis generates a series of expressed genes encoding various neurotransmission components. Using targeted RNAi, the authors test the contribution of these components to the locomotor activity rhythms. They reveal that the slowpoke potassium channel plays a role in the PI cells to generate robust activity rhythms in constant conditions. Slowpoke was already known to be involved in the control of the circadian behavior but only clock cells were reported to be a site for slowpoke clock function, and the present study indicates that at least part of the non-clock cell function takes place in the PI.

The molecular and behavioral data are clearly presented and provide interesting information about slowpoke role in the clock neuron downstream circuit, which remains poorly understood.

My only comment is about the very limited description of the cell isolation procedure. I think that the authors should provide more information on how they isolate single cells for RNAseq analysis.

We apologize for the lack of detail. We have now extended our methods section to provide more information about how we isolated single cells for RNA seq:

“We used a fine-tipped glass micropipette for cell harvesting. The micropipette was inserted into a pipet holder and connected by flexible tubing to a 1 mL syringe. Using a micromanipulator, we slowly advanced the micropipette towards the PI region. To avoid collecting cellular debris while advancing through brain tissue, we maintained light positive pressure by blowing through the syringe. Once the micropipette was just touching the soma of the cell of interest, we applied gentle mouth suction until the cell entered the pipet tip. We then broke off the tip containing the harvested cell into a 1.7 mL microcentrifuge tube and immediately processed the contents for antisense RNA amplification.” (Lines 144-151).

In addition, in response to a point raised by Reviewer #1, we have added detail explaining our use of kurs58-GAL4 and C767-GAL4, along with Dilp2mCherry, to isolate non-DILP-expressing PI cells for harvesting:

“We used a single-cell transcriptional profiling approach to identify potential circadian output genes expressed by relevant PI cell populations. The PI is comprised of ~30 cells, but only specific subsets have been implicated in control of rest:activity rhythms. Because the 14 DILP-expressing PI cells do not appear to contribute to rest:activity regulation (4,5), we sought to target non-DILP-expressing PI cells for single-cell sequencing following GFP-guided cell capture. To identify the cells of interest, we drove GFP expression with either of two GAL4 lines, kurs58-GAL4 or C767-GAL4, which are both active in non-DILP-expressing PI cells (5). Notably, constitutive neuronal activation under the control of either kurs58-GAL4 or C767-GAL4 compromises rest:activity rhythm strength, confirming the relevance of these cells (5). The flies used for single-cell capture also included a Dilp2mCherry construct, which selectively labels the DILP-expressing PI cells. This served two purposes: first, Dilp2mCherry acted as a landmark to aid in PI localization; second, it allowed us avoid selecting DILP-expressing cells, which could be easily identified based on their mCherry fluorescence (see Fig 1A-B).” (Lines 119-131).

---

## [Editor Report · Decision Letter 1]

4 Mar 2021

PONE-D-20-37778R1

Slowpoke functions in circadian output cells to regulate rest:activity rhythms

PLOS ONE

Dear Dr. Cavanaugh,

Thank you for submitting your manuscript to PLOS ONE. After careful consideration, we feel that it has merit but does not fully meet PLOS ONE’s publication criteria as it currently stands. Therefore, we invite you to submit a revised version of the manuscript that addresses the points raised during the review process.

We look forward to receiving your revised manuscript.

Kind regards,

Nicholas Simon Foulkes, D.Phil

Academic Editor

PLOS ONE

Journal Requirements:

Additional Editor Comments (if provided):

The revised version of the manuscript has improved considerably and all the reviewers issues have been addressed convincingly. However, I would still hold the authors to include a simple diagram/cartoon, as originally requested by Reviewer 1, that would help the non-specialist reader follow the Drosophila brain architecture and the experimental approach.

---

## [Author Response · Author response to Decision Letter 1]

4 Mar 2021

Based on the feedback of the academic editor (and initial input from Reviewer 1), we now include a diagram of the fly brain detailing the neurochemical classes of PI neurons. This has been added as Figure 1B. We hope that this aids the non-specialist reader to understand the Drosophila brain architecture and the specifics of our experimental approach.

---

## [Editor Report · Decision Letter 2]

15 Mar 2021

Slowpoke functions in circadian output cells to regulate rest:activity rhythms

PONE-D-20-37778R2

Dear Dr. Cavanaugh,

We’re pleased to inform you that your manuscript has been judged scientifically suitable for publication and will be formally accepted for publication once it meets all outstanding technical requirements.

Kind regards,

Nicholas Simon Foulkes, D.Phil

Academic Editor

PLOS ONE
---

## [Editor Report · Acceptance letter]

17 Mar 2021

PONE-D-20-37778R2 

*Slowpoke* functions in circadian output cells to regulate rest:activity rhythms 

Dear Dr. Cavanaugh:

I'm pleased to inform you that your manuscript has been deemed suitable for publication in PLOS ONE. Congratulations! Your manuscript is now with our production department. 

Kind regards, 

on behalf of

Dr. Nicholas Simon Foulkes 

Academic Editor

PLOS ONE